# MSEDDI: Multi-Scale Embedding for Predicting Drug—Drug Interaction Events

**DOI:** 10.3390/ijms24054500

**Published:** 2023-02-24

**Authors:** Liyi Yu, Zhaochun Xu, Meiling Cheng, Weizhong Lin, Wangren Qiu, Xuan Xiao

**Affiliations:** Department of Computer, School of Information Engineering, Jingdezhen Ceramic University, Jingdezhen 333403, China

**Keywords:** drug—drug interaction, knowledge graph, graph neural network, self-attention mechanism

## Abstract

A norm in modern medicine is to prescribe polypharmacy to treat disease. The core concern with the co-administration of drugs is that it may produce adverse drug—drug interaction (DDI), which can cause unexpected bodily injury. Therefore, it is essential to identify potential DDI. Most existing methods in silico only judge whether two drugs interact, ignoring the importance of interaction events to study the mechanism implied in combination drugs. In this work, we propose a deep learning framework named MSEDDI that comprehensively considers multi-scale embedding representations of the drug for predicting drug—drug interaction events. In MSEDDI, we design three-channel networks to process biomedical network-based knowledge graph embedding, SMILES sequence-based notation embedding, and molecular graph-based chemical structure embedding, respectively. Finally, we fuse three heterogeneous features from channel outputs through a self-attention mechanism and feed them to the linear layer predictor. In the experimental section, we evaluate the performance of all methods on two different prediction tasks on two datasets. The results show that MSEDDI outperforms other state-of-the-art baselines. Moreover, we also reveal the stable performance of our model in a broader sample set via case studies.

## 1. Introduction

With the rapid increase in approved drugs, it has become common for physicians to treat patients with polypharmacy [1]. In clinical practice, the co-administration of multiple drugs magnifies the complexity of treatment management and may produce unexpected drug—drug interactions (DDIs) and even be life-threatening [2]. In order to safely administer the drug, it is necessary to assist in the determination of DDI by in vivo and in vitro trials. However, the experimental approach is very time-consuming and expensive; many potential interplays are challenging to detect early in drug development [3].

As a result, people have turned their attention to computational methods, hoping to build algorithmic models to predict possible drug interaction events. Compared with traditional wet assays, computer-based methods enable efficient screening of potential DDI. These methods are roughly divided into four categories: literature extraction-based, matrix factorization-based, ensemble learning-based, and network-based.

Literature-based extraction methods utilize natural language processing techniques to extract drug—drug interactions from biomedical literature. With the proliferation of biomedical literature, a large amount of drug knowledge is hidden in unstructured literature [4,5,6]. Consequently, the automatic extraction model is designed to detect DDI from textual information. Sun et al. [7] developed a deep convolutional neural network with multiple layers and small convolutions, namely DCNN, for DDI extraction. In further study [8], they introduced a Recurrent Hybrid Convolutional Neural Network (RHCNN) containing canonical convolution and dilated convolution. However, traditional CNNs cannot solve the problem of long sentences because they only deal with adjacent words and ignore the long-distance dependencies between words. Kavuluru et al. [9] proposed a character-level Recurrent Neural Network (char-RNN) to accomplish the DDI extraction task. Despite the promising results of these approaches, the task usually required high-quality human annotations, which is very time-consuming.

The DDI network can be transformed into an adjacency matrix representing the inter-drug associations. Neural Collaborative Filtering (NCF) [10] canceled the simple linear inner product operation of the traditional matrix factorization (MF) and designed a neural network-based collaborative filtering method to capture the complex structure in the network data. Yu et al. [11] deemed the identification of potential side effects of drugs as a matrix reconstruction using a linear neural network layer. Shi et al. [12] presented a triple matrix factorization-based method named TMFUF. Furthermore, some new matrix factorization models were implemented based on manifold learning and neural networks [13]. Zhu et al. [14] designed a dependent network to model drug dependency and proposed an attribute supervised learning model probabilistic dependent matrix tri-factorization (PDMTF) for adverse drug—drug interaction (ADDI) prediction. Yu et al. [15] developed a novel method DDINMF based on semi-nonnegative matrix factorization. Zhang et al. [16] proposed the manifold regularized matrix factorization method for DDI prediction.

The ensemble learning-based method combines several models to achieve better performances than individual models. Deepika and Geetha [17] adopted a semi-supervised learning framework with network representation learning and meta-learning from four drug datasets to predict DDIs. Zhang et al. [18] proposed the sparse feature learning ensemble method with linear neighborhood regularization(SFLLN) for the DDI prediction. Zhu et al. [19] proposed a unified multi-attribute discriminative representation learning (MADRL) model for ADDI prediction, which considered the intrinsic connection and extrinsic difference between drugs. AttentionDDI [20] is a Siamese self-attention multimodal neural network for DDI prediction that integrates multiple drug similarity measures obtained from the comparison of drug features, including drug targets, pathways, and gene expression profiles.

The network-based approach is to mine drug potential information from drug-related networks or graphs. With the help of the cheminformatics toolkit RDKit [21], the drug SMILES can be represented as a molecular chemical structure graph. Hu et al. [22] developed various molecular learning methods based on pre-trained GNNs for extracting the chemical structure embedding of drug molecules. Chen et al. [23] employed a pre-trained message-passing neural network to generate structural representations of drugs, where the raw material only involves the number and chirality of atoms and the type and orientation of bonds. Nyamabo et al. [24] created a gated message passing neural network (GMPNN-CS) which learns chemical substructures with different sizes and shapes from the molecular graph representations of drugs for DDI prediction between a pair of drugs. GNN-MolGNet [25], a powerful molecular graph model, was pre-trained at the node level and graph level, which can capture valuable chemical insights to produce interpretable representations. In the rich biomedical network consisting of various entities, Qian et al. [26] built a gradient boosting-based classifier using feature similarity and feature selection methods to speed up the procedure and achieve robust prediction performance. LR-GNN [27] defined a propagation rule that captures the node embeddings of each GCN [28] encoding layer to construct link representations (LRs). GCNMK [29] noted that types of DDI are associated with diverse mechanisms. Two graph convolution kernels are constructed by “increasing” the related DDI network and “decreasing” the related DDI network. Knowledge Graph (KG) [30,31] is an effective tool for representing entity relationships, which integrates multiple data sources to build a rich biomedical information repository. To analyze graph data, graph embedding methods (e.g., TransE [32], ComplEx [33]) learn low-dimensional dense vectors for both relations and nodes and feed them into downstream biomedical tasks [34,35].

In general, several existing methods aim only to predict whether two drugs interact, and do not delve into the specific meaning of drug interaction and changes in metabolism levels from the perspective of human metabolism. As a result, more and more DDI prediction methods are now focusing on interaction events. Ryu et al. [36] expressed drug interaction events as 86 metabolic events in the form of human-readable sentences. They proposed a deep learning model DeepDDI based on drug chemical structural information to predict specific DDI events. Deng et al. [13] proposed a multimodal deep learning framework, DDIMDL, which combines diverse drug features to predict 65 DDI-related events. Lin et al. [37] screened a dataset with 100 DDI events from the DrugBank database. Up to now, most studies have been set in the transductive setting, and they have been limited to predicting other possible DDI linkages between drugs with known mechanisms of action, i.e., training and test sets share a portion of the drug. Nevertheless, such methods ignore the inductive setting (a.k.a. cold-start scenarios) [13,24,37,38], all unique drugs in DDI pairs are divided into two drug sets, and then DDI pairs are grouped into training set and testing set according to the belonging of drugs in pairs.

In this work, we investigate a DDI multi-classification method applicable to a more challenging inductive setting. To this end, we propose a multi-scale embedding and multi-structure neural network model, named MSEDDI, which learns and fuses multi-source drug features to ensure good performance in the cold-start task. We design a multi-channel feature learning strategy that can separately process knowledge graph embeddings from biomedical networks, notation embeddings from SMILES strings, and chemical structure embeddings from molecular graphs. These features involve both the rich biological association information external to the drug and the chemical and structural information inherent to the drug. We construct different neural network channels according to the data structure characteristics of various features to obtain more effective drug representation. MSEDDI utilizes a self-attention mechanism to fuse abstracted features from each channel and feeds the fused features into a fully connected layer to predict drug interaction events. Experimental results show that MSEDDI achieves the best performance in the cold-start scenario, where DDI prediction can be made on drugs that the model was not trained on.

## 2. Results and Discussion

### 2.1. Experiment Setup and Evaluation

In this work, we are more concerned with new drugs not in the known DDI network, that is, drugs that have never been involved in model training. There are two goals for the DDI prediction task: the interactions between known drugs and new drugs (Task 1) and the interactions between new drugs (Task 2). These are known as the cold-start scenario. Firstly, we randomly divided the drug set into five subsets with 5-fold cross-validation: 4 as the known drugs and the rest as the new drugs. Then the original DDI dataset is automatically divided into a known—known drug pair set, a known—new drug pair set, and new—new drug pair set according to the drug division. The model is trained on the known DDI dataset and implements Task 1 and Task 2 on known—new DDIs and new—new DDIs, respectively.

For evaluation, we adopt four metrics to measure the multi-classification performance of models, including accuracy (ACC), area under the precision-recall-curve (AUPR), area under the ROC curve (AUC), and F1, where AUPR and F1 are more sensitive for greatly unbalanced datasets. Furthermore, we calculate AUC and AUPR with the “micro” mode, which considers each element of the label indicator matrix as a label. In contrast, F1 with “macro” mode calculating for each label and then find their unweighted mean.

### 2.2. Hyperparameter Search

The tuning of hyperparameters is to satisfy that the model is adequately trained while avoiding overfitting as much as possible to achieve the best performance. We investigate six hyperparameters that affect MSEDDI performance: loss weight, dropout, batch size, epoch, hidden layer dimension, and learning rate. The model performance metrics under different profiles are depicted in Figure 1, and the performance change curve of the model under various configurations is observed. Therefore, we set the loss weight to 1 × 10^−5^, dropout to 0.2, batch size to 512, epoch to 350, hidden layer dimension to 64, and learning rate to 1 × 10^-4^.

For the prediction task on Dataset 2, we fine-tune the aforementioned partial hyperparameters’ search range to accommodate the dataset’s expansion. To better fit larger dataset, we increase the hidden layer dimension to raise the model complexity and decrease the loss weight. The hyperparameter search process is illustrated in Appendix A. The specific hyperparameter settings of our model on both datasets are viewed in Appendix A.

### 2.3. Method Comparison

To highlight the excellent performance of our model, we compare MSEDDI with the following state-of-the-art DDI prediction methods:
MDF-SA-DDI [37] combines a pair of drugs in four ways and inputs the combined drug feature representation into four different drug fusion networks (Siamese network, convolutional neural network, and two auto-encoders). Then, transformer blocks perform latent feature fusion.DDIMDL [13] constructs deep neural network-based sub-models by using four drug features: chemical substructures, targets, enzymes, and pathways, and then adopts a joint DNN framework to combine the sub-models and learn cross-modality representations of drug pairs to predict DDI events.Lee’s method [39] uses autoencoders and a deep feed-forward network that are trained using the structural similarity profiles, Gene Ontology term similarity profiles, and target gene similarity profiles of known drug pairs to predict the pharmacological effects of DDIs.DeepDDI [36] consists of the structural similarity profile (SSP) generation pipeline and deep neural network (DNN). The two SSPs of each input drug structure in pair are combined and fed into the DNN to predict the interaction type between drugs.


The performance of our model and the baselines listed above on Dataset 1 is presented in Table 1. From all metric results, we conclude that MSEDDI outperforms the suboptimal method in Task 1 by 0.58% on ACC, 4.2% on AUPR, and 3.88% on AUC. The improvement of MSEDDI in Task 2 is 0.73% on ACC, 1.89% on AUPR, and 8.68% on AUC. The differential performance of all methods in two tasks demonstrates that the lack of prior knowledge brings a great test to the predictive ability of the model. On Dataset 2, we still compare these state-of-the-art methods in two target tasks. Table 2 indicates that our model achieves the best performance. For Task 1, MSEDDI outperforms the runner-up by up to 10.33% on ACC, 14.95% on AUPR, 0.85% on AUC, and 5.67% on F1. For Task 2, MSEDDI improves by 15.15% on ACC, 21.46% on AUPR, 1.77% on AUC, and 1.74% on F1.

Relatively, the performance of all models on Dataset 2 is saliently improved, especially for our method, indicating that the increase in sample size can help improve the performance of deep learning models. For Task 1, the ACC, AUPR, and F1 of MSEDDI on Dataset 2 are more 10% higher than that of Dataset 1. For Task 2, the AUPR and F1 are, respectively, almost boosted by 26% and 15%. In terms of F1, the score of our model in Dataset 1 is not ideal but achieves the best on the larger Dataset 2. These occasions illustrate that MSEDDI can play a better predictive ability when dealing with the sheer size of samples.

Furthermore, we utilize AUPR and AUC calculated in Task 1 to investigate the performance of partial outstanding models in each event. The AUPR and AUC of all prediction models for each event on two datasets is shown in Figure 2 and Appendix A. The original AUPR and AUC of each event on two datasets are listed in Appendix A. It can be observed that MSEDDI occupies half of the best scores on AUPR for the top 10 events with higher frequency on Dataset 1. While on the other minor events, our model has no commendable points. For AUPR on Dataset 2, MSEDDI has a notable superiority on more than 80% of events. Compared to the 65 events in Dataset 1, our model has different degrees of performance improvement in 65 corresponding events out of 100 events in Dataset 2. In addition, we count and detect a gap ranging from a few times to more than a dozen times in the number of DDI for each event between both datasets. This also reflects that our model can exert the potential advantages for major events and large datasets. To further analyze the comprehensive performance of diverse methods, we display the statistical boxplots of AUPR and AUC for partial baselines across all events in Figure 3 and Appendix A. It should be noted that in Dataset 1, MSEDDI performs better on the top several major events with a higher sample proportion and worse in the remaining dozens of minor events with a smaller percentage. Nevertheless, MDF-SA-DDI turns out to be the opposite. Therefore, the AUPR and AUC averaged by sample size of MSEDDI are the highest in Table 1, while MDF-SA-DDI is the best from the event perspective in these boxplots.

In summary, DeepDDI with the single similarity feature and DNN performs the worst. DDIMDL and Lee’s method have their merits in terms of AUPR and F1. The similarity between the two methods is that they exploit four different similarities as drug features and input into the predictor. MDF-SA-DDI establishes a multi-structure neural network to re-learning the similarity features adopted by DDIMDL and applies the self-attention mechanism to merge the processed features. As opposed to drug similarity, our approach extracts drug features from knowledge graphs constructed from vast amounts of biomedical data. In addition, we derive information on the graph structure and sequence structure from the intrinsic chemical structure of the drug, ensuring the diversity of drug representations. According to the traits of various features, we constructed three deep neural network channels for processing the raw features. Overall, our method markedly outperforms other advanced methods and exhibits better predictive performance.

### 2.4. Ablation Study

To illustrate the effectiveness of the features in our model and the rationality of the structural design, we conduct ablation studies with three feature variants and three structural variants, which are described as follows:
MSEDDI_NC only retains the knowledge graph embedded kge and network channel module in MSEDDI.MSEDDI_SC relearns SMILES notation embedding sne using only sequence channel network.MSEDDI_GC only uses the chemical structure embedding cse of drug molecule and graph channel module.MSEDDI_NO_ADD removes the residual connection trick in the sequence channel of MSEDDI.MSEDDI_NO_CONV removes the convolution module of the graph channel on the basis of MSEDDI.MSEDDI_NO_ATTEN simply considers the fusion of various features learned from the representation learning module by splicing before DDI prediction.


The performance of MSEDDI and its six variants in two tasks on Dataset 1 is presented in Appendix A. As far as the feature variants are concerned, MSEDDI_NC is the best, and each metric value is relatively close to MSEDDI, while MSEDDI_GC performs the worst. These indicate that knowledge graph embedding extracted from the biomedical network is the most compelling feature. Moreover, the fusion of different structural features facilitates the performance of the model. Regarding the model structure, MSEDDI_NO_ADD has no significant performance degradation, and MSEDDI_NO_ATTEN is the most variable. From the results of the structural variants, the residual connection contributes the least to MSEDDI, and the attention mechanism used to fuse features has the most apparent impact on MSEDDI, which also shows that there are specific differences in the quality of the three features. Moreover, Appendix A shows the results of the models and variants on Dataset 2. We come to the same conclusion that knowledge graph embedding is the best-performing feature, the effect of the residual connection is negligible, and the attention mechanism is still an indispensable way for feature fusion.

Overall, the final morphology of our model achieves the best performance compared to all variants, which attests that the diverse fusion of features is crucial for the model to cope with extremely challenging tasks. Moreover, designing a reasonable and effective neural network for representation re-learning according to the characteristics of the features can further improve the quality of drug representation and contribute to the downstream prediction phase.

### 2.5. Representation Visualization

In this section, we perform a visualized analysis for the final representations of drug pairs obtained by learning and fusing three heterogeneous features in an end-to-end manner. For display purpose, we chose Dataset 1 with smaller sample size and number of DDI events, and visualized the drug pair representation drawn from predictor MLP output in Task 1.

The drug pair representations are projected and visualized in the 2D space by uniform manifold approximation and projection (UMAP). As shown in Appendix A, the color bar on the right indicates the colors of 65 DDI events. Each colored dot represents a drug pair and the DDI event to which it belongs, and most of the dots are concentrated in several clusters. In particular, the top-ranked large sample size DDI categories are highly dense. The visualization results show that MSEDDI can learn high-quality drug representations for accurately identifying DDI events.

### 2.6. Case Study

In the epilogue of this work, we validate the practical capabilities of MSEDDI with the help of a case study. Referring to the approach of Deng et al., we train our model with all DDIs in Dataset 1 derived from DrugBank and then predict the remaining drug pairs. We pay attention to five events with the highest frequencies and check the top 20 predictions related to each event. We used the Interaction Checker tool provided by DrugBank to validate these predictions.

Seventy-four DDI events can be confirmed among 100 events. For example, the interaction between Lapatinib and Mestranol is predicted to cause event #0, which means the metabolism of Mestranol can be decreased when combined with Lapatinib. The interaction between Mesoridazine and Fospropofol is predicted to cause event #1, which means the risk or severity of adverse effects can be increased when Mesoridazine is combined with Fospropofol. More detailed supporting materials about confirmed drug—drug interaction events are provided in Appendix A.

In addition, we also find that a certain drug may be closely related to a certain DDI event. For example, the metabolism of 8 drugs of the 18 DDIs related to event #0 are reduced owing to Cholecalciferol. Ten of the 19 verified DDIs related to increasing the risk or severity of adverse effects are associated with Meprobamate. Against the training set, we count 64 reverse-order drug—drug pairs (DDIs), which refers to the fact that the model that learns the input order of di and dj is still able to recognize the correct labels of dj and di, which shows the robustness of MSEDDI.

## 3. Materials and Methods

### 3.1. Dataset

There are two benchmark datasets for our study, which were constructed by Deng et al. [13] and Lin et al. [37]. Deng et al. extracted and screened 572 drugs and 37,264 pairwise DDIs (Dataset 1) from DrugBank. Each DDI is represented as a four-tuple structure: (drug A, drug B, mechanism, action), where the “mechanism” means the effect of the drug on human metabolism, and the “action” stands for an increase or decrease in metabolic level. The combination of “mechanism&action” constitutes 65 different interaction events, and the interaction of the drug pair corresponds to only one event. Lin et al. applied a similar approach to select a more extensive dataset from DrugBank, including 1258 drugs with 323,539 pairwise DDIs associated with 100 events (Dataset 2). The percentages of all events for both datasets are shown in Figure 4. All DDI event profiles for both datasets are provided in Appendix A.

Drug Repurposing Knowledge Graph (DRKG) [40] is a comprehensive biological knowledge graph relating genes, compounds, diseases, biological processes, side effects, and symptoms. It includes 97,238 entities belonging to 13 entity types and 5,874,261 triplets belonging to 107 edge-types.

### 3.2. Problem Formulation

In our study, the dataset is a DDI network composed of drugs and interaction events. We present the DDI network as $G=\{(d_{i},l_{k},d_{j})|d_{i},d_{j}\in D,l_{k}\in L,i,j\in [1,N_{d}],k\in [1,N_{l}]\}$, where D denotes drug set, L represents interaction event set, and $N_{d}$, $N_{l}$ are the magnitude of D and L, respectively. In terms of drug representation, the drug set corresponds to molecular structure graph set $M=\left\{m_{1},m_{2},\ldots ,m_{N_{d}}\right\}$ and SMILES sequence set $S=\left\{s_{1},s_{2},\ldots ,s_{N_{d}}\right\}$. In addition, we also represent drugs through the Drug Repurposing Knowledge Graph (DRKG) K={(h,r,t)|h,t∈E,r∈R}, where E denotes the entity set (e.g., drug, protein, disease, etc.), and R represents the relation set (e.g., interaction, similarity, etc.). Based on the drug representation and DDI network G, MSEDDI aims to learn a multi-class prediction algorithm f:D×D→L to output the probability of a specific event $l_{k}$ of a drug pair $\left(d_{i},d_{j}\right)$. Subsequently, we elaborate MSEDDI in detail.

### 3.3. Overview of MSEDDI

The MSEDDI framework structure is presented in Figure 5. Our model consists of three modules, namely drug representation, representation learning, and feature fusion. The drug’s rudimentary representations are derived from three feature embedding methods. The node embedding method extracts the knowledge graph embedding kge from the biomedical network. The word embedding method converts the SMILES string into notation embedding sne. The graph embedding method learns chemical structure embedding cse from the drug molecular graph. Next, three-channel neural networks connected in parallel form the representation learning module for feature processing. In the network channel, we employ multi-layer linear networks to regularize the feature dimension of the knowledge graph embedding. In the sequence channel, we design the network structure of a 1D convolutional layer splice sequence encoder for learning the local and global contexts of the SMILES sequence. In the graph channel, we process chemical structure embeddings generated by multiple GNN methods using linear networks and a 1D convolutional neural network. The feature fusion module stacks the embeddings learned from the above channels and fuses them by leveraging a multi-headed self-attentive mechanism. In the end, all channel features are flattened and fed into a linear layer predictor to determine the interaction event of drug pairs. Subsequently, we elaborate MSEDDI in detail.

### 3.4. Drug Representation

#### 3.4.1. Knowledge Graph Embedding

For the biomedical network DRKG, we adopt the knowledge graph embedding method TransE [32] to learn the embedding representation of entities and relations. TransE is a representative translational distance model that represents entities and relations as vectors in the same semantic space. In terms of vector computation, it could mean adding a head h to a relation r should approximate to the relation’s tail t, that is h+r≈t. For the negative set, the model applies a variation of negative sampling by corrupting triplets (h,r,t). Specifically, TransE corrupt h or t to produce a negative triplet $\left(h^{\prime },r,t\right)$ or $\left(h,r,t^{\prime }\right)$, and the premise of sampling is that either $h^{\prime }$ and t or h and $t^{\prime }$ are unrelated. Therefore, the positive set $D^{+}$ and the negative set $D^{-}$ are represented as follows:(1)$$\begin{array}{c} D^{+}=\left\{\left(h,r,t\right)\right\}\\ D^{-}=\left\{\left(h^{\prime },r,t\right)\cup \left(h,r,t^{\prime }\right)|h^{\prime },t^{\prime }\in E\right\} \end{array}$$

TransE model is trained to minimize the margin-based loss function which is given by:(2)ℓkge=∑D+∑D−max(0,γ−f(D+)+f(D−))
where γ denotes the margin parameter, the scoring function f is negative distance between h+r and t (i.e., −‖h+r−t‖). TransE runs iteratively on DRKG to reduce the ℓkge and update the representation of all entities and relations. In other words, all representations conform to the complex topology of the knowledge graph to the maximum extent. Similarly, we can obtain the reasonable representation of the specific drug. Finally, we screen out drug entity di∈D and its knowledge graph embedding Hikge∈ℝ1×dim_kge.

#### 3.4.2. SMILES Notation Embedding

The simplified molecular-input line entry system (SMILES) is a specification in the form of a line notation for describing the structure of chemical species using short ASCII string. For each drug’s SMILES sequence si∈S, we employ the word2vec [41] model to encode the notation in the sequence to derive the feature tensor zi. In simple terms, each word is signified by the mean of a fixed range of context word vectors and fed into two linear transformation layers to yield a probability vector zi (predicted probability) corresponding to each notation. The objective loss function ℓsne of the word2vec model is to calculate the error between the true label zi and the predicted probability zi′:(3)zi′(j)=Softmax(mean(∑q∈C(j)zi(q)W1)W2)
(4)ℓsne=−∑i=1Nd∑j=1len(i)∑k=1vzi(j,k)logzi′(j,k)
where C(j) represents other notations within the context of the *j*th notation, zi(j)∈ℝ1×v stands for the one-hot vector of the *j*th notation in the SMILES sequence, v represents the one-hot vector dimension of notation, that is, the number of all notations, W1∈ℝv×u,W2∈ℝu×v are both trainable weight matrices, dim_sne is the vector dimension of notation after training (u≪v), len(i) represents the SMILES sequence length of drug di, and zi(j,k) refers to the *k*th component value of the jth notation vector. Finally, the weight W1 obtained by training the model by minimizing ℓsne is used to generate the sequence embedding Hisne for drug di:(5)Hisne=zi×W1
where Hisne∈ℝlen(i)×u. The length of SMILES strings is indeterminate, more than 90% of which are less than 100. Being aware of this issue, we fold the feature tensors with lengths greater than 100 into lengths less than 100 and average them, and then unify the lengths of all feature tensors by padding zero vectors. The sequence embedding is mark as Hisne∈ℝ100×dim_sne.

#### 3.4.3. Chemical Structure Embedding 

For each drug $d_{i}$, we leverage the open-source cheminformatics library RDKit [21] to generate a molecular structure graph $m_{i}\in M$ according to SMILES string, and $m_{i}=\left(\nu ,\varepsilon \right)$ where ν represents atoms, ε represents chemical bonds between atoms, and they both attach abundant physicochemical information.

In this work, we adopt eight atomic attributes [38] and four chemical bond attributes [11] (i.e., type, conjugated, ring, stereo) to describe molecular graphs. Upon these facts, the graph embedding learning method (e.g., MPNN [42], Weave [43], AttentionFP [44]) is employed to extract feature from the molecular structure graph. The process involves a message passing phase and a readout phase. In the message passing phase, the aggregation function At is employed to fuse the information of other nodes and edges within the fixed neighborhood of each node, and the fusion information is used to refresh the node’s content through the update function Ut. Therefore, message passing can be described as below:(6)axt+1=∑yϵN(x)At(hxt, hyt, exy)
(7)hxt+1=Ut(hxt,axt+1)
where t is the iteration round, x denotes a node in the graph mi, N(x) represents the adjacent nodes of node x, hxt stands for the intermediate state of node x at time t, exy indicates the attributes of edge between x and y. In particular, At and Ut are inherently linear layers with the same dimension weight matrix to deliver information between nodes. Finally, eigenvector Hi of graph mi is calculated by readout function on all node representation at t step. The readout phrase can be formulated as:(8)Hi*=readout(hxt|x∈v), *∈{mpnn, weave, afp}
where Hi*∈ℝ1×dim_cse. It should be pointed out that there are certain differences between MPNN, Weave and AttentiveFP in the two phases. The message passing phase is basically a linear network, but differs in depth, structure, dimension, and composition. In the readout phase, MPNN adopts the Set2Set [45], which includes the LSTM network. Weave simply in tandem with a linear layer in the tail, and AFP builds three layers of global pooling modules.

### 3.5. Representation Learning Module

For the sake of mapping various features extracted by the aforementioned methods to the identical dimensional space, MSEDDI sets up a multi-layer perceptron module MLP in multiple parts, consisting of two linear layers, a batch normalization [46] layer, and a non-linear activation function. The multi-layer perceptron is depicted as:(9)$$MLP=LeakyReLU\left(BN1d\left(H_{in}W_{1}\right)\right)W_{2}$$
where $H_{in}$ represents the input tensor, $W_{1}$ and $W_{2}$ are the trainable weights for the two linear layers, respectively. LeakyReLU denotes a leaky version of Rectified Linear Unit, BN1d refers to one-dimensional batch normalization, which re-normalizes the vectors on the corresponding dimension using the mean and standard-deviation calculated per dimension over the mini-batches.

#### 3.5.1. Network Channel

We apply the MLP to re-learn the knowledge graph embedding Hkge, and then stack the representation of drug di and drug dj as the network channel feature Pijkge of drug pair (di,dj). The process is described as follows:(10)Pijkge=|MLP(Hikge)MLP(Hikge)|
where Pijkge∈ℝ2×dim, dim represents the hidden layer dimension, and the outermost | | means vertical stacking of vectors.

#### 3.5.2. Sequence Channel

Initially, we splice the SMILES notation embeddings Hsne of two drugs (e.g., di and dj) with interaction along the sequence direction as drug pair representation. Next, MSEDDI utilizes one-dimensional convolution to pay attention to sequence substructures and extract local information of sequence. In order to capture the global information, we exploit the self-attention mechanism [47] to calculate the attention score among each notation, and aggregate and update the sequence context information of the notation. Subsequently, the layer normalization [48] module re-normalizes the notation embedding vectors to move the data into the effective region of the downstream activation function. The feed-forward layer consists of two linear layers with weights Wl1 and Wl2 and an activation layer GELU. It maps data to high-dimensional space and then to low-dimensional space to learn more abstract features and enhance the expressive ability of upstream features. Finally, to avoid the gradient problem, we add a residual connection [49] between the convolutional layer and the feed-forward layer, which does an element-wise sum of the front and rear outputs as sequence channel feature Pijsne. The channel structure can be marked as:(11)Pijs1=LeakyReLU(MaxPool1d(BN1d(Conv1d(Hisne⊕Hjsne)), 4))
(12)Pijs2=LN(Atten(Pijs1))
(13)Pijsne=Pijs1+LN(GELU(Pijs2Wl1)Wl2)
where Pijsne∈ℝ50×dim, Pijs1∈ℝ(100×2÷4)×dim, Wl1∈ℝdim×2dim, Wl2∈ℝ2dim×dim, LN denotes layer normalization.

#### 3.5.3. Graph Channel

We feed the three chemical structure embeddings H* into three independent MLP for unification into a hidden dimension dim, and the outputs are stacked as features for drug pair (di,dj). In the convolution layer, the kernel gradually performs convolution operations along the stacking direction to fuse the drug representations generated from different embedding methods and output the graph channel feature Pijcse. The process of the graph channel is represented as follows:(14)Pij*=|MLP*(Hi*)MLP*(Hj*)|
(15)Pijcse=LeakyReLU(BN1d(Conv1d(|PijmpnnPijweavePijafp|)))
where Pijcse∈ℝ(2×3)×dim, *∈[mpnn, weave,afp], MLP* corresponds to three MLPs that handle graph embedding individually.

### 3.6. Feature Fusion Module

In this module, MSEDDI combines all features of the drug pair and predicts the DDI event. We stack the high-order features learned from the above three channels into a drug pair representation Pij, including the network channel representation Pijkge, the sequence channel representation Pijsne, and the graph channel representation Pijcse. To better fuse these features, we input the drug pair representation Pij into a self-attention layer to improve the expressiveness of each embedding. For the downstream DDI prediction task, MSEDDI flattens the final feature of the drug pair and feeds it to the predictor MLP, which maps the drug pair representation to Nl-label vector space. The process can be defined as follows:(16)Pij=|PijkgePijsnePijcse|
(17)y=MLP(Flatten(LN(Atten(Pij))))
where y∈ℝ1×Nl, Flatten means data flattening operation.

## 4. Conclusions

In this work, we propose a novel deep learning framework named MSEDDI for multi-class DDI prediction tasks. MSEDDI adopts three heterogeneous embeddings to represent drugs, derived from the knowledge graph of network structure, the SMILES string of sequence structure, and the molecular chemical structure of graph structure. We build three channel networks to relearn the features and fuse them by the attention mechanism. Compared with other advanced methods, our model is significantly superior in evaluation metrics. In addition, we verify the soundness of features and structures based on model variants and visualize the predicted drug pair representations (as described in Appendix A). Finally, we perform the case studies to identify the DDI events not included in our dataset.

There exists a limitation in our work that the knowledge graph embedding technique is not suitable for a more extreme case, which is new drugs isolated from known biomedical networks. This restriction exists for other similarity-based or association-based methods as well. In future work, we will focus on mining information about the drug itself to develop effective drug representation methods and investigate the DDI prediction task even further in the stricter cold-start scenario. 

## Figures and Tables

**Figure 1 ijms-24-04500-f001:**
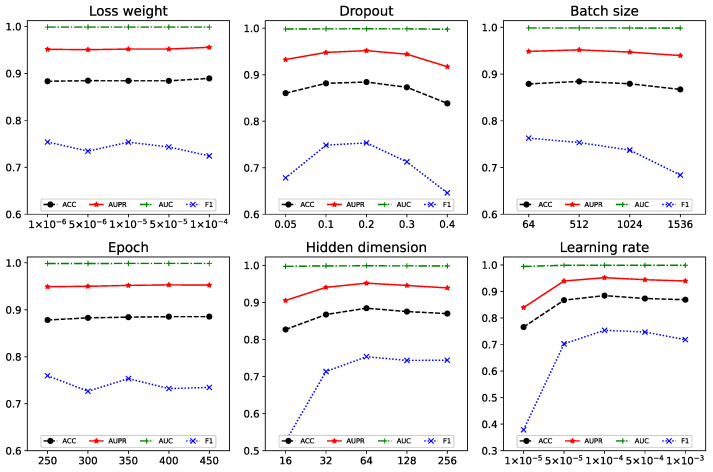
The metric scores under different hyperparameter settings on Dataset 1.

**Figure 2 ijms-24-04500-f002:**
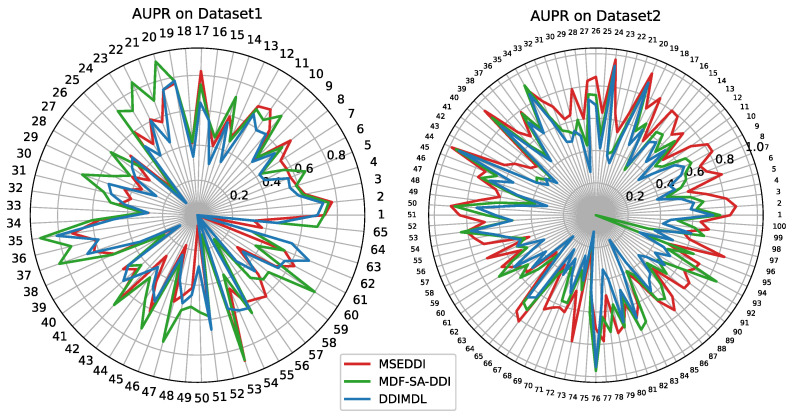
The AUPR of partial baselines for each event.

**Figure 3 ijms-24-04500-f003:**
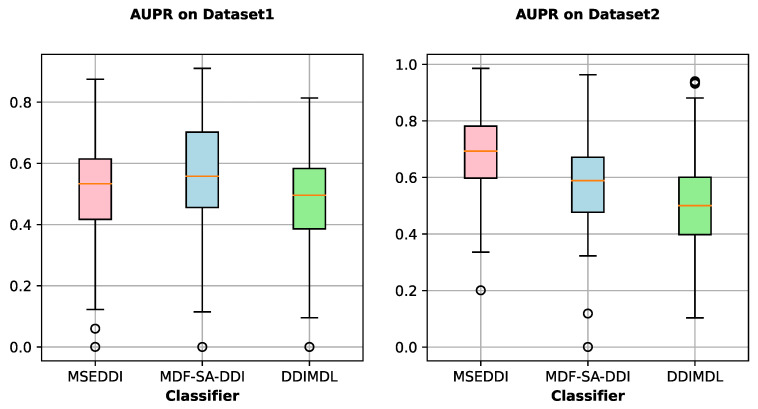
The statistic of AUPR for partial baselines in all events.

**Figure 4 ijms-24-04500-f004:**
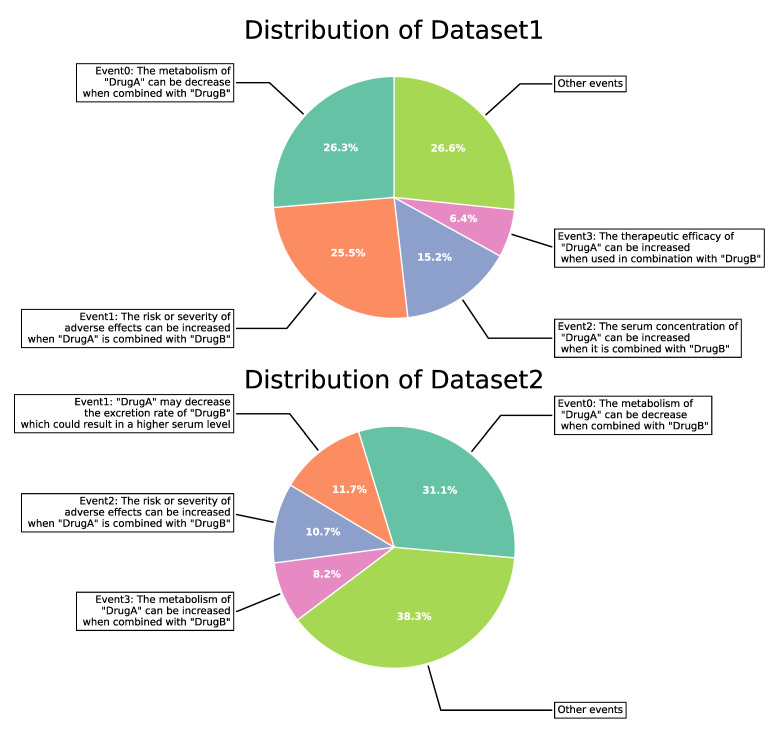
The statistics of DDI event.

**Figure 5 ijms-24-04500-f005:**
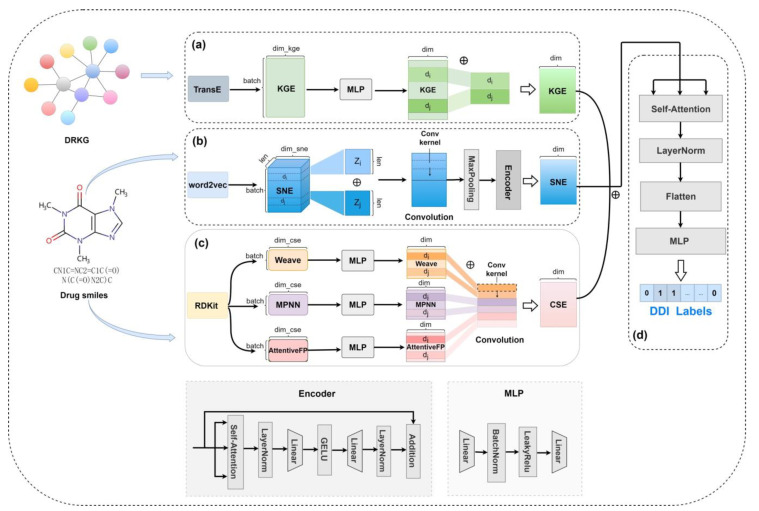
The entire framework of MSEDDI. (**a**) Network channel. MSEDDI applies the graph embedding algorithm TransE to mine the relationship and topological information between drugs and other biomedical identifiers from the biomedical network DRKG as the drug knowledge graph embedding kge. The kge is spliced into the network representation of drug pairs through MLP. (**b**) Sequence channel. We employ the word2vec algorithm to convert the drug SMILES string into SMILES notation embedding sne. MSEDDI splices drug pair sne and processes them into the sequence representation using a convolutional network and encoder module for learning the local and global contexts of the SMILES sequence. (**c**) Graph channel. We utilize three graph neural network algorithms to extract the chemical structure embedding cse of the drug molecular graph acquired by the RDKit tool. Subsequently, MSEDDI sets three MLPs and a convolutional network for generating the graph representation of drug pairs. (**d**) Feature fusion. MSEDDI stacks three highly abstracted drug pair representations to feed into the self-attention module and MLP for downstream DDI prediction tasks.

**Table 1 ijms-24-04500-t001:** The performance of all methods on Dataset 1.

Method	Task 1				Task 2			
	ACC	AUPR	AUC	F1	ACC	AUPR	AUC	F1
DeepDDI	0.5774	0.5594	0.9575	0.3416	0.3602	0.2781	0.9059	0.1373
Lee’s method	0.6405	0.6244	0.9247	0.5039	0.4097	0.3184	0.8302	0.2022
DDIMDL	0.6415	0.6558	0.9799	0.4460	0.4075	0.3635	0.9512	0.1590
MDF-SA-DDI	0.6459	0.6390	0.9435	0.5471	0.4378	0.3810	0.8675	0.2326
**MSEDDI**	**0.6517**	**0.6810**	**0.9823**	**0.4771**	**0.4451**	**0.3999**	**0.9543**	**0.1691**

The best results are highlighted in boldface.

**Table 2 ijms-24-04500-t002:** The performance of all methods on Dataset 2.

Method	Task 1				Task 2			
	ACC	AUPR	AUC	F1	ACC	AUPR	AUC	F1
DeepDDI	0.5883	0.5851	0.9746	0.4709	0.3611	0.2820	0.9264	0.1868
Lee’s method	0.6917	0.7119	0.9687	0.5934	0.4867	0.4349	0.9093	0.3082
DDIMDL	0.6720	0.7086	0.9885	0.5817	0.4699	0.4386	0.9685	0.3032
MDF-SA-DDI	0.6664	0.6820	0.9862	0.5919	0.4794	0.4450	0.9686	0.2937
**MSEDDI**	**0.7697**	**0.8315**	**0.9947**	**0.6486**	**0.6309**	**0.6596**	**0.9863**	**0.3111**

The best results are highlighted in boldface.

## Data Availability

The source code and data are available at http://github.com/yuliyi/MSEDDI (accessed on 15 January 2023).

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
