# Peer review of "MSEDDI: Multi-Scale Embedding for Predicting Drug—Drug Interaction Events"

_ijms, 2023, doi:10.3390/ijms24054500_

Round 1

Reviewer 1 Report

The paper is well written and of interest. Topic,  multi-scale embedding for predicting drug-drug interaction events. This topic is building computational models for drug development. There are no major spelling/grammar errors included 5 figures, 17 equations and 1 table.

In order to safely administer the drug, it is necessary to assist in the determination of DDI by in vivo and in vitro trials. However, the experimental approach is very time-consuming and expensive, many potential interplays are challenging to detect early in drug development.

The auhors have investigated a DDI multi-classification method applicable and applied representation visualization

Conclusion - In this work, the authors propose a novel deep learning framework named MSEDDI for multi-class DDI prediction tasks. MSEDDI adopts three heterogeneous embeddings to represent drugs, derived from the knowledge graph of network structure, the SMILES string of sequence structure, and the molecular chemical structure of graph structure. We build three channel networks to relearn the features and fuse them by the attention mechanism.

It can be accepted for publication as is. 

Author Response

We very much appreciate Reviewer #1’s overall positive and professional comments.

Reviewer 2 Report

Dear Author,

The study you have discussed and the methods used in the study are very well thought out and organized. As a researcher working on these issues, it is wonderful that the approaches used in the solution and the gaps in the current literature are very well-identified and made into a feasible and user-friendly application. Congratulations.

Author Response

We very much appreciate Reviewer #2’s overall positive and professional comments.

Reviewer 3 Report

Yu and coauthors report the development and implementation of a new machine learning based multi-scale embedding drug-drug interaction (DDI) prediction tool: MSEDDI. In particular, the authors designed three-channel networks to process biomedical network-based knowledge graph embedding, SMILES sequence-based notation embedding, and molecular graph-based chemical structure embedding, respectively whose features are then fused and then passed through a self-attention, layernorm, flattening and multilayer perception to predict DDI.

The overall study is of interest to the AI and ML based drug discovery and interaction predictions. The feature representation, extraction and prediction structure designs are technically sound, however there are concerns regarding the results and conclusions and comments listed below for consideration.

(1) Page 8, section 2.6. Case Study. "We used the Interactions Checker tool provided by drugs.com to validate these predictions. Seventy-four DDI events can be confirmed among 100 events. For example, the interaction between Methoxyflurane and Cholecalciferol is predicted to cause event #0, which 303 means the metabolism of Methoxyflurane can be decreased when combined with Cholecalciferol. The interaction between Meprobamate and Ketazolam is predicted to cause event #1, which means the risk or severity of adverse effects can be increased when Meprobamate is combined with Ketazolam."

It appears that the examples of drug-drug interactions claimed by the authors in the Case Study are not confirmed at the Interactions Checker tool. Methoxyflurane and Cholecalciferol are suggested to have no evidence support when they are looked up using the Interaction Checker tool. Ketazolam can not be looked up by the Interaction Checker tool (unless the authors somehow used another name of the same drug, which needs clarification). Unfortunately with these discrepancy, the conclusions are in doubt. Unless there are other systematic way to support the current predictions, I suggest use a list of examples from left-out test set (i.e. not used in training) to support current method's soundness.

(2) Page 10, Figure 5 (b) After the Conv Kernel and Max pooling, how the sequence order of two drugs of interest di, dj (Drug A and Drug B) can be preserved or it actually doesn't matter di and dj or dj and di when it comes to current method of drug-drug interaction? The reason of this comment is that it appears the predicted DDI event labels indicate different types of events for example, according to Figure 4 on page 9: Event #0 is defined as The metabolism of "Drug A" can be decrease when combined with "Drug B", which implies "Drug A" effect on "Drug B" may not necessarily be the same as "Drug B" effect on Drug A"

(3) Figure 5 (a). It's somewhat unclear what the source DRKG input and TransE generated feature KGE looks like. Can the authors give a concrete example to explain? 

Author Response

Response to Reviewer 3 Comments

Point 1: Page 8, section 2.6. Case Study. "We used the Interactions Checker tool provided by drugs.com to validate these predictions. Seventy-four DDI events can be confirmed among 100 events. For example, the interaction between Methoxyflurane and Cholecalciferol is predicted to cause event #0, which 303 means the metabolism of Methoxyflurane can be decreased when combined with Cholecalciferol. The interaction between Meprobamate and Ketazolam is predicted to cause event #1, which means the risk or severity of adverse effects can be increased when Meprobamate is combined with Ketazolam."

It appears that the examples of drug-drug interactions claimed by the authors in the Case Study are not confirmed at the Interactions Checker tool. Methoxyflurane and Cholecalciferol are suggested to have no evidence support when they are looked up using the Interaction Checker tool. Ketazolam can not be looked up by the Interaction Checker tool (unless the authors somehow used another name of the same drug, which needs clarification). Unfortunately with these discrepancy, the conclusions are in doubt. Unless there are other systematic way to support the current predictions, I suggest use a list of examples from left-out test set (i.e. not used in training) to support current method's soundness.

Response 1: We are very sorry, but it is not clear what caused your verification to fail. Here we have provided the address of the Interactions Checker tool. Alternatively, you can access the DrugBank database to verify the DDI. Both ways are valid and are linked below:

https://dev.drugbank.com/demo/ddi_checker

https://go.drugbank.com/

Point 2: Page 10, Figure 5 (b) After the Conv Kernel and Max pooling, how the sequence order of two drugs of interest di, dj (Drug A and Drug B) can be preserved or it actually doesn't matter di and dj or dj and di when it comes to current method of drug-drug interaction? The reason of this comment is that it appears the predicted DDI event labels indicate different types of events for example, according to Figure 4 on page 9: Event #0 is defined as The metabolism of "Drug A" can be decrease when combined with "Drug B", which implies "Drug A" effect on "Drug B" may not necessarily be the same as "Drug B" effect on Drug A"

Response 2: Good point. There is no order for the two drugs in DDI. The corresponding DDI events are the same whether they are di and dj or dj and di. During the construction of the dataset, we only took one of the pairs to avoid duplicate samples. Moreover, the Interactions Checker tool and the DrugBank database do not distinguish the order between drugs.

Point 3: Figure 5 (a). It's somewhat unclear what the source DRKG input and TransE generated feature KGE looks like. Can the authors give a concrete example to explain?

Response 3: Thanks for your good suggestion. We have added some explanations in the subsection '3.4.1. Knowledge graph embedding '. However, we are awfully sorry and not sure what you mean about the example. An example of the feature generation process with a specific drug? Here, we give the generation method of KGE for reference.

https://github.com/gnn4dr/DRKG/blob/master/embedding_analysis/Train_embeddings.ipynb

Round 2

Reviewer 3 Report

The authors have provided additional information to address my concerns. I have a few comments to their responses for consideration.

(1) Point 1. Page 8, 2.6. Case study. "We used the Interactions Checker tool provided by drugs.com to validate these predictions". 

The response at least addresses the concern of validity of the predictions. However, the authors shall correctly cite the resource tool they used to verify DDI. It appears the authors intended to use drugbank tool instead of drugs.com tool as written in their manuscript, the latter doesn't show those verifications the authors claimed..

(2) Point 2. The authors responded that "Response 2: Good point. There is no order for the two drugs in DDI. The corresponding DDI events are the same whether they are di and dj or dj and di. During the construction of the dataset, we only took one of the pairs to avoid duplicate samples. Moreover, the Interactions Checker tool and the DrugBank database do not distinguish the order between drugs."

Drugbank DDI checker indeed can take input of two drugs no matter what's the order the end user typed in, and output a DDI that specifies the order and type of the interactions. This only means that the DrugBank DDI tool has already been programmed to accept the di and dj no matter what the order of the input. I suggest for the current new method MSEDDI to be robust and correctly predict DDI independent of the input order of di and dj, the training process shall include dj and di as well as di and dj as input even if they should have exactly the same labels of output. In principle this shall allow the algorithm to be more robust because the embedding of current MSEDDI method is indeed taking the input layer in a sequential order. It's not uncommon to use Data Augmentation to improve ML algorithm. Alternatively, perhaps it's worth checking if the authors shall be able to verify the current method can successfully predict both input in order di and dj as well as reverse input order dj and di (for the case studies..)

Author Response

Response to Reviewer 3 Comments

Point 1: Page 8, 2.6. Case study. "We used the Interactions Checker tool provided by drugs.com to validate these predictions".

The response at least addresses the concern of validity of the predictions. However, the authors shall correctly cite the resource tool they used to verify DDI. It appears the authors intended to use drugbank tool instead of drugs.com tool as written in their manuscript, the latter doesn't show those verifications the authors claimed..

Response 1: Thanks for your good suggestion. We have modified the first paragraph in the section '2.6. Case study'.

Point 2: The authors responded that "Response 2: Good point. There is no order for the two drugs in DDI. The corresponding DDI events are the same whether they are di and dj or dj and di. During the construction of the dataset, we only took one of the pairs to avoid duplicate samples. Moreover, the Interactions Checker tool and the DrugBank database do not distinguish the order between drugs."

Drugbank DDI checker indeed can take input of two drugs no matter what's the order the end user typed in, and output a DDI that specifies the order and type of the interactions. This only means that the DrugBank DDI tool has already been programmed to accept the di and dj no matter what the order of the input. I suggest for the current new method MSEDDI to be robust and correctly predict DDI independent of the input order of di and dj, the training process shall include dj and di as well as di and dj as input even if they should have exactly the same labels of output. In principle this shall allow the algorithm to be more robust because the embedding of current MSEDDI method is indeed taking the input layer in a sequential order. It's not uncommon to use Data Augmentation to improve ML algorithm. Alternatively, perhaps it's worth checking if the authors shall be able to verify the current method can successfully predict both input in order di and dj as well as reverse input order dj and di (for the case studies..)

Response 2: Good point. We re-run the case study and compare the predictions with the training set, and detect 64 reverse-order drug-drug pairs out of 74 validated DDIs. We have modified the content of the section '2.6. Case study '.